# N-Acetylcysteine and Aripiprazole Improve Social Behavior and Cognition and Modulate Brain BDNF Levels in a Rat Model of Schizophrenia

**DOI:** 10.3390/ijms23042125

**Published:** 2022-02-15

**Authors:** Zofia Rogóż, Kinga Kamińska, Marta Anna Lech, Elżbieta Lorenc-Koci

**Affiliations:** 1Department of Pharmacology, Maj Institute of Pharmacology, Polish Academy of Sciences, 12 Smętna Street, 31-343 Kraków, Poland; rogoz@if-pan.krakow.pl (Z.R.); k.kamin@if-pan.krakow.pl (K.K.); hereta@if-pan.krakow.pl (M.A.L.); 2Department of Neuro-Psychopharmacology, Maj Institute of Pharmacology, Polish Academy of Sciences, 12 Smętna Street, 31-343 Kraków, Poland

**Keywords:** neurodevelopmental model of schizophrenia, schizophrenia-like symptoms, levels of BDNF mRNA and its protein, aripiprazole, N-acetylcysteine

## Abstract

Treatment of negative symptoms and cognitive disorders in patients with schizophrenia is still a serious clinical problem. The aim of our study was to compare the efficacy of chronic administration of the atypical antipsychotic drug aripiprazole (7-{4-[4-(2,3-dichlorophenyl)-1-piperazinyl] butoxy}-3,4-dihydro-2(1H)-quinolinone; ARI) and the well-known antioxidant N-acetylcysteine (NAC) both in alleviating schizophrenia-like social and cognitive deficits and in reducing the decreases in the levels of the brain-derived neurotrophic factor (BDNF) in the prefrontal cortex (PFC) and hippocampus (HIP) of adult Sprague-Dawley rats, that have been induced by chronic administration of the model compound L-buthionine-(S, R)-sulfoximine (BSO) during the early postnatal development (p5–p16). ARI was administered at doses of 0.1 and 0.3 mg/kg while NAC at doses of 10 and 30 mg/kg, alone or in combination. Administration of higher doses of ARI or NAC alone, or co-treatment with lower, ineffective doses of these drugs significantly improved social and cognitive performance as assessed in behavioral tests. Both doses of NAC and 0.3 mg/kg of ARI increased the expression of BDNF mRNA in the PFC, while all doses of these drugs and their combinations enhanced the levels of BDNF protein in this brain structure. In the HIP, only 0,3 mg/kg ARI increased the levels of both BDNF mRNA and its protein. These data show that in the rat BSO-induced neurodevelopmental model of schizophrenia, ARI and NAC differently modulated BDNF levels in the PFC and HIP.

## 1. Introduction

Among the heterogeneous group of psychotic disorders [1], schizophrenia is distinguished by a wide range of psychopathological features, such as positive symptoms (hallucinations, delusions, impaired thinking, disorganized behavior), negative symptoms (lack of motivation, social functions deficits, flattening of affect) and cognitive dysfunction (attention, memory and executive functions deficits) [2,3,4,5].

Pharmacological treatment of schizophrenia became available thanks to the development of chlorpromazine, and since then, antipsychotic drugs, have been widely used until today. These drugs are effective against positive symptoms, including hallucinations and delusions, but have no or little beneficial effect on the negative or cognitive symptoms [5,6]. Moreover, the effects of antipsychotics are associated with significant side effects and not all patients respond positively to their application. So, given that both negative and cognitive symptoms remain largely untouched by current antipsychotic drugs, treating schizophrenia still constitutes a serious clinical problem and challenges researches to seek more effective therapies [7]. To achieve these goals, a thorough understanding of the biological basis of schizophrenia is essential.

The etiology of schizophrenia is still unknown, but in accordance to the prevailing hypothesis, it is considered a neurodevelopmental disorder caused by multiple interactions of genetic and environmental factors [8,9,10]. These interactions are initiated during embryonic or early postnatal development, leading to structural and functional brain abnormalities, some of which appear in middle to late adolescence and are fully expressed in adulthood [11,12,13]. Hence, adolescence has been proposed as a critical period of vulnerability to the development of schizophrenia [14,15,16]. The study of the living human brain during development is limited for ethical and practical reasons. Therefore, animal models of schizophrenia are an important tool for investigating the pathophysiology and treatment of this disease [17]. Genetic engineering or pharmacological manipulations are applied to create animal models of schizophrenia that represent at least some of the behavioral, biochemical and structural changes that characterize the disease [18,19]. In terms of pharmacological tools, various model substances, such as phencyclidine, ketamine, lipopolysaccharide (LPS) and methylazoymethanol (MAM) are administered to adult or developing rodents during the embryonic or early postnatal period to induce these changes [18,19,20].

Additionally, for many years the dysregulation of glutathione (GSH) synthesis has been postulated as an important factor in the pathophysiology of schizophrenia [21,22,23,24,25,26,27]. Consistently with the view above, L-buthionine-(S, R)-sulfoximine (BSO), a specific inhibitor of the key enzyme in GSH biosynthesis [28,29], i.e., γ-glutamate cysteine ligase (GCL), may serve as a model compound to induce the schizophrenia-like changes in rats. In terms of behavioral alterations, a study by Cabungcal et al. [30] showed that chronic treatment of Osteogenic Disorder Shionogi (ODS) mutant rats or Wistar rats with BSO during the early postnatal development (p5–p16) resulted in impairment of some cognitive function assessed in adulthood in the radial maze with controlled olfactory cues. Using the same BSO administration regimen, our research performed on Sprague-Dawley rats [31,32] revealed that in addition to cognitive deficits assessed in the novel object recognition test (NOR), which appeared already in mid adolescence and persisted into adulthood, BSO treatment led to the emergence of social deficits assessed in the social interaction test (SIT). However, such treatment with BSO did not induce changes in rats’ behavior that would resemble positive symptoms in patients with schizophrenia [31,32]. Furthermore, BSO-treated rats did not exhibit in adulthood a specific feature of schizophrenia patients [32] i.e., increased sensitivity to the amphetamine-induced excess of dopamine [33,34,35].

Regarding the biochemical changes induced by repeated administration of BSO in the early postnatal period, in our previous study performed in 16-day-old pups, 4 h after the last dose of BSO, drastic decreases in GSH levels and significant increases in the sulfur amino acid methionine (MET) concentrations were found in peripheral tissues (liver, kidneys) [31]. In the brain of 16-day-old pups after the last BSO dose, there was a slight but significant decrease in the GSH content in the prefrontal cortex (PFC) and no changes in its level in the HIP, but both structures showed a significant increase in cysteine (Cys) concentration. Moreover, in these rats, a significant decrease in the Met level in the PFC was found with a simultaneous significant increase in its concentration in the HIP [31]. At the molecular level, chronic treatment with BSO resulted in a significant decrease in the global methylation of DNA in the PFC and no changes in the HIP in 16-day-old pups [31]. On the other hand, such chronic treatment with BSO during early postnatal development led in adult 93-day-old rats to a significant increase in the global methylation of DNA in the PFC and its reduction in the HIP [31]. The latter effects indicate that BSO is a model compound that disrupts the physiological epigenetic status in the PFC and HIP areas during early postnatal development and later in adulthood, which may be associated with functional implications. It is worth mentioning that epigenetic regulation plays an important role in the brain development, synaptic plasticity and formation of long-term memory [36,37,38,39].

The neurotrophic hypothesis of schizophrenia [40] postulates that pathological changes in the brain of schizophrenia patients are a consequence of progressive disturbances in the processes involving trophic factors. Post-mortem studies of schizophrenia patients’ brains revealed alterations in BDNF in certain brain regions [41,42,43]. In particular, some studies have shown that BDNF mRNA and its protein levels were reduced in the frontal cortex and the HIP in the brains of these patients compared to healthy controls [44,45,46], while others reported that BDNF expression increased in both of these structures [47,48]. Some animal models of schizophrenia confirm the down-regulation of BDNF levels observed in the post-mortem studies of brains of schizophrenic patients. Thus, in a neurodevelopmental model of schizophrenia, e.g., in rats with a neonatal ibotenic lesion of the ventral HIP [49,50] or in rats exposed to the MAM toxin during embryogenesis [19], decreases in BDNF mRNA and its protein were observed in both the PFC and HIP in adulthood [51,52,53]. Also our recently published study showed that repeated administration of BSO in the early postnatal period resulted in significant decreases in the levels of BDNF mRNA and its protein both in the PFC and HIP in adulthood [31]. Moreover, several preclinical studies in rats revealed that antipsychotic drugs could modulate the levels of BDNF in the cerebral cortex and HIP. Most of these studies have shown that atypical antipsychotics induce BDNF up-regulation, while typical antipsychotics, such as haloperidol reduce or do not affect BDNF levels [54,55,56,57]. The above data clearly suggest that changes in BDNF levels are associated both with pathology and treatment of schizophrenia.

The aim of the current study was to compare the therapeutic efficacy of the atypical antipsychotic drug aripiprazole, 7-{4-[4-(2,3-dichlorophenyl)-1-piperazinyl] butoxy}-3,4-dihydro-2(1H)-quinolinone (ARI) and the antioxidant N-acetylcysteine (NAC) highly recommended for the treatment of schizophrenia [58,59], administered alone or in combination, in reversing the social and cognitive deficits caused by chronic BSO given during early postnatal days (p5–p16). To relate the BSO-induced declines in BDNF mRNA and its protein in the PFC and HIP to the therapeutic effects of ARI and NAC in ameliorating schizophrenia-like deficits, both markers were determined 24 h after the last chronic doses of these drugs. To address all these issues, ARI at doses of 0.1 and 0.3 mg/kg, as in our previous studies [60,61], and NAC at doses of 10 and 30 mg/kg, determined on the basis of the conversion of clinical doses of NAC used in patients with schizophrenia according to the average weight of an adult rat [59], were administered chronically on postnatal days p68–p91.

## 2. Results

### 2.1. Chronic Treatment of Adult Sprague-Dawley Rats with ARI or NAC Reverses the Social Deficits Induced by BSO in Early Postnatal Development

Social behavior of adult Sprague-Dawley rats receiving chronic ARI or NAC, either alone or in combination, was assessed 24 h after their last doses using two parameters, i.e., the total time spent by two rats in social interactions and the number of these interactions (Figure 1A,B).

A one-way ANOVA for the planned comparisons performed for the total time spent in social interactions (F_(9,50)_ = 26.990, *p* < 0.00001) showed that repeated administration of BSO to neonatal rats on the postnatal days p5–p16, resulted in a significant reduction in the total time spent in social interactions after these rats reached adulthood (p90). Chronic treatment with ARI (0.3 mg/kg) or NAC (30 mg/kg) alone in the postnatal days p68–p90 almost completely reversed the BSO-mediated effect (Figure 1A). Lower doses of the tested drugs were ineffective. This analysis also revealed that the combined administration of ineffective doses of ARI (0.1 mg/kg) and NAC (10 mg/kg) significantly increased the total time spent by two rats in social interactions when compared to the effects of the tested doses of these drugs administered alone. Moreover, the combined treatment with ineffective dose of ARI (0.1 mg/kg) with effective dose of NAC (30 mg/kg) caused further increases in the total time spent in social interactions compared the effects of these drugs doses administered alone. Also the combined administration of the effective dose of ARI (0.3 mg/kg) with an ineffective or effective dose of NAC significantly extended the total duration of social interaction between the two tested rats compared to values of this parameter in the corresponding groups of rats receiving the same doses of these drugs separately (Figure 1A).

As with the total time of social interactions, the one-way ANOVA for the planned comparisons carried out for the number of social interactions (F(9,50) = 15.795, *p* < 0.0001) showed that as a consequence of the chronic BSO administration to neonatal rats on the postnatal days (p5–p16) there was a significant reduction in the number of these interactions in adulthood. As shown in Figure 1B, the efficacy of the tested doses of ARI (0.1 and 0.3 mg/kg) and NAC (10 and 30 mg/kg) in reversing the effects of BSO as well as the impact of different combinations of these doses of drugs on the number of social interactions was the same as described above with respect to the total duration of social interactions.

### 2.2. Chronic Treatment of Adult Sprague-Dawley Rats with ARI or NAC Reverses the Cognitive Deficits Induced by BSO in Early Postnatal Development

The novel object recognition test (NOR) designed to assess cognitive impairments in rats was performed in all studied groups on the next day after SIT (Figure 2).

During the acquisition trial (session T1), all rats representing particular groups spent equal time exploring two identical objects (Figure 2A). In the retention trial (session T2), rats from the control group administered chronically vehicle and from BSO group, chronically treated with ARI at a dose of 0.3 mg/kg or NAC at a dose of 30 mg/kg alone, and also receiving combinations of these drugs: (ARI 0.1 mg/kg × NAC 10 mg/kg), (ARI 0.1 mg/kg × NAC 30 mg/kg), (ARI 0.3 mg/kg × NAC 10 mg/kg) and (ARI 0.3 mg/kg × NAC 30 mg/kg) explored the novel object significantly longer than the familiar one (Figure 2B).

A one-way ANOVA for the planned comparisons performed for the recognition index in all studied groups (F(9,70) = 40.10, *p* < 0.0001) showed that BSO administered in early postnatal life caused a long-lasting reduction in its value in adulthood, but higher doses of ARI or NAC administered chronically alone between days p68–p91 after birth reversed the values of this index to the level of the control group (Figure 2C). Interestingly, also the combined administration of ineffective doses of ARI (0.1 mg/kg) and NAC (10 mg/kg) significantly reversed the BSO-mediated cognitive deficits evaluated as the recognition index. Moreover, the combined administration of an ineffective dose of ARI (0.1 mg/kg) with the effective dose of NAC (30 mg/kg) or the effective dose of ARI (0.3 mg/kg) with the ineffective dose of NAC (10 mg/kg) also increased the values of the recognition indexes compared to the value of these indexes in groups receiving either ARI (0.1 mg/kg) or NAC (10 mg/kg) alone, respectively (Figure 2C).

### 2.3. Chronic Treatment of Adult Sprague-Dawley Rats with ARI and/or NAC Modulates the Changes in BDNF mRNA and Protein Levels in the Prefrontal Cortex and Hippocampus Induced by BSO in the Early Postnatal Life

A one-way ANOVA for the planned comparisons performed for the prefrontal cortex (PFC) of adult 91-day-old rats, receiving chronically BSO between postnatal days p5–p16, showed that both BDNF mRNA (F_(7,56)_ = 9748, *p* < 0.001)) and its protein (F_(7,56)_ = 3466, *p* < 0.005) levels were significantly reduced when compared to the level of these parameters in the group of 91-day-old control rats that were administered vehicle during early postnatal development (Figure 3A,B).

In the PFC, chronic administration of ARI (0.3 mg/kg) or NAC at both doses tested (10 or 30 mg/kg) to adult rats (68 to 90 days of age) pretreated with BSO in the early postnatal development (p5–p16) resulted in significant increases in the BDNF mRNA expression compared to its expression in the PFC in the group of adult BSO pretreated rats receiving chronically vehicle (Figure 3A). However, chronic administration of ARI alone at a dose of 0.1 mg/kg and a combination of ARI (0.1 mg/kg) with NAC at doses of 10 or 30 mg/kg did not change the BSO-induced decline in the BDNF mRNA expression in the PFC. In addition, BDNF mRNA levels in groups treated with the combinations of ARI (0.1 mg/kg) + NAC (10 mg/kg) or ARI (0.1 mg/kg) + NAC (30 mg/kg) were significantly lower than in groups which were administered 10 or 30 mg/kg of NAC alone, respectively. Contrary to the differentiated influence of ARI and NAC on the BDNF mRNA expression in the PFC, all examined drug doses and their combinations significantly increased the level of BDNF protein in this brain structure compared to the level of BDNF protein in the BSO group receiving in adulthood (p68–p90) chronically vehicle (Figure 3B).

Like in the PFC, a one-way ANOVA for the planned comparisons performed for the hippocampus (HIP) of 91-day-old rats, chronically receiving BSO between postnatal days p5–p16, demonstrated that both BDNF mRNA (F(7,56) = 21,173, *p* < 0.001) and its protein (F(7,56) = 3229, *p* < 0.01) levels were significantly reduced when compared to the corresponding levels of these parameters in the group of 91-day-old control rats, which were administered vehicle in the early postnatal period (Figure 4A,B).

In this brain structure, only chronic administration of ARI at a dose of 0.3 mg/kg resulted in significant increases in both BDNF mRNA and its protein levels, while other doses of the tested drugs and their combinations were ineffective. Interestingly, the combined administration of ARI (0.1 mg/kg) + NAC (10 mg/kg) resulted in further decrease in BDNF mRNA compared to the BSO-treated groups receiving vehicle or NAC (10 mg/kg) while the combined administration of ARI (0.1 mg/kg) + NAC (30 mg/kg) reduced the level of BDNF mRNA only compared to the vehicle-treated BSO group (Figure 4A). However, these drops in BDNF mRNA levels did not result in further decreases in BDNF protein levels in these groups (Figure 4B).

## 3. Discussion

The present study showed that the atypical antipsychotic drug aripiprazole (ARI) and the well-known antioxidant N-acetylcysteine (NAC), administered chronically alone to adult Sprague-Dawley rats on the postnatal days p68–p91 at doses of 0.3 and 30 mg/kg, respectively, equally reversed the social and cognitive deficits that were induced by chronic treatment with the model compound BSO in the early postnatal development (p5–p16). Lower doses of ARI (0.1 mg/kg) and NAC (10 mg/kg) were ineffective in reducing social and cognitive deficits, but their co-administration successfully reversed them to the level of the control group. The latter effect indicates the existence of positive interactions between the tested doses of ARI and NAC, and, at the same time, suggests that there must be some common mechanism underlying this interaction that leads to the improvement in social behavior and cognitive functions. Further analysis of other ARI and NAC dose variants given in the combined administration paradigm seems to indicate their more powerful effect on social behavior than on memory impairment. Interestingly, co-administration of 0.1 mg/kg ARI with 30 mg/kg NAC had the strongest improving effect on social functioning among all the dose variants tested. On the other hand, all variants of the combined administration of ARI and NAC reversed BSO-induced cognitive deficits, as assessed by the recognition indexes, only to the control group level. It is worth recalling that ARI is a novel atypical antipsychotic which has a high affinity for dopamine D_2_ and serotonin 5-HT1A receptors [62,63,64], and can also act as a partial agonist for these receptors [65,66]. Moreover, it is a potent partial dopamine D_3_ and D_4_ receptor agonist and serotonin 5-HT_2A_ receptor antagonist [64,67]. Findings from preclinical studies suggest that pro-cognitive effects of ARI are due to its partial agonism at dopaminergic receptors [68]. This effect of ARI, however, does not seem to be sufficient to explain the particularly beneficial effect of this drug in the reduction of negative symptoms, especially when it is administered in combination with NAC. In clinical trials, improvement of cognitive functions does not always lead to an amelioration of social functioning. Therefore, it can be assumed that the alleviation of negative symptoms in the BSO-induced rat model of schizophrenia by chronic administration of ARI and NAC may be associated to a greater extent with the modulation of other neurotransmitter systems, e.g., glutamate transmission rather than dopamine transmission. In mutiple clinical trials, ARI was effective in reducing positive and negative symptoms in schizophrenia patients, but only partially improved cognitive functions [63,69]. In our previous studies in Sprague-Dawley rats, ARI at a dose of 0.3 mg/kg, the same as used in the present study, was effective in reversing the MK-801-induced both social and cognitive deficits as assessed by the NOR and SIT tests, respectively [60,61]. The latter studies also showed that co-administration of an ineffective dose of ARI (0.1 mg/kg) with ineffective doses of antidepressants, such as escitalopram (5 mg/kg) or mirtazapine (5 mg/kg), alleviated social and cognitive deficits induced by MK-801, what confirms the involvement of non-dopamine neurotransmitter systems in modulating these symptoms by ARI. Other studies have also shown that ARI mitigates schizophrenia-like behavior and inhibits glutamate release induced by NMDA receptor antagonists [70,71].

Along with the improvement in social behavior and cognition, ARI at a dose of 0.3 mg/kg and both tested doses of NAC significantly increased the levels of BDNF mRNA in the PFC, which were reduced by BSO pre-treatment during early postnatal life. As for the impact of these drugs on BDNF protein levels, both doses of ARI and NAC as well as the combination of a lower dose of ARI with NAC increased the levels of this protein in the PFC. In contrast to the PFC, only 0.3 mg/kg of ARI increased the levels of BDNF mRNA and its protein in the HIP. The data described above show that beneficial effects of ARI and NAC on BDNF levels in the BSO-induced rat model of schizophrenia were mainly restricted to the PFC. Since BDNF is an important signaling molecule that is essential for learning and memory processes [49,72], it is likely that an increase in the level of its protein in the PFC may have some implications for the improvement of cognitive functions.

As mentioned in the chapter Introduction, the treatment with BSO during early postnatal life did not induce behavioral changes in adult rats corresponding to positive symptoms in patients with schizophrenia [31,32], therefore it was reasonable to suppose that the occurrence of social deficits and cognitive impairment in these rats could be associated rather with more severe disturbances of glutamatergic than dopaminergic transmission [73]. The glutamatergic hypothesis of schizophrenia, as an alternative to the dopaminergic one, is based on studies in healthy humans showing that NMDA receptor antagonists, such as ketamine and phencyclidine, transiently induced features characteristic of schizophrenia, including psychosis, negative symptoms and cognitive impairments [73,74,75]. For a long time, it has been thought that the neurotransmitter glutamate pool, accounting for 50–60% of the total glutamate pool, derives from glutamine-glutamate shuttle between neurons and glia, while a much smaller amount of glutamate transmitter is produced by the glycolysis and tricarboxylic acid cycle [76,77,78]. However, since after inhibition of the glutamine-glutamate exchange mechanism, a quick replenishment of glutamate neurotransmission is still possible [79], it has been suspected that there is another endogenous glutamate storage buffer. More recent studies strongly suggest that GSH may serve as a physiological reservoir of glutamate [80,81]. In support of this view, these studies showed that BSO, which blocks the utilization of glutamate for GSH synthesis when added to the primary cortical neurons in culture, increased the total glutamate levels by 25–50%, depending of the BSO dose used. Relating the above data to the model of schizophrenia used in our study, it seems that the increase in glutamate levels in the rat brain caused by chronic BSO treatment during postnatal days p5–p16, which coincides with the developmental switch in the action of GABA through GABAA receptor from excitatory to inhibitory [82], may evoke compensatory changes in the excitation-inhibition balance in adulthood. Therefore, it is debatable whether the transient inhibition of GSH synthesis by BSO during early postnatal development leads to hypo- or hyperfunction of glutamatergic transmission in specific brain structures of adult Sprague-Dawley rats, especially in the PFC, in which both an increase and a decrease in glutamatergic signaling [83,84] is associated with the emergence of the schizophrenia-like social and cognitive deficits.

Assuming that glutamatergic transmission dysfunction underlies the changes observed in the BSO-induced rat model of schizophrenia, in this part of discussion we present potential mechanisms indicating that the therapeutic effects of ARI and NAC in alleviating social and cognitive deficits may be related to modulating just this transmission. As to therapeutic efficacy of NAC in patients with schizophrenia, in some clinical trials this drug was efficient against the negative symptoms and cognitive impairment, but failed to reduce the positive symptoms [85,86,87]. Other clinical studies using NAC have shown improvements in the cognitive domains of processing speed that has been associated with negative symptoms [88], and working memory [89]. Regarding the mode of NAC action in patients with schizophrenia, Yolland et al. [59], based on numerous literature data, concluded that the improvement of cognitive functions after administration of NAC might result from a combination of various factors, including reduction of oxidative stress and neuroprotection of cognitive networks. Moreover, recent clinical studies suggest that activation of the cystine-glutamate antiporter by systemic NAC administration alleviated the cognitive impairment in schizophrenia and ketamine-induced psychosis [90]. On the other hand, preclinical studies have shown that NAC inhibits glutamate release in the mPFC induced by systemic administration of phencyclidine [91], although the cystine-glutamate antiporter, located on the cell membrane of astrocytes, releases glutamate through the exchange of extracellular cystine and intracellular glutamate [92]. Thus, it is likely that the inhibitory effect of NAC on the increased glutamate release in the mPFC induced by systemic administration of NMDA receptor antagonists may be regulated by another mechanism. Since NAC increases the activity of group II metabotropic glutamate receptors [93], it seems that these receptors are responsible for the inhibitory effect of NAC on glutamate release in the mPFC. In the most recent preclinical study, Fukuyana et al. [94] tried to identify both the mechanism responsible for MK-801-induced increase in glutamate release in the medial prefrontal cortex (mPFC), and to define the mechanisms responsible for normalization of this release after ARI or NAC treatment. Comparative analysis of the systemic administration of MK-801 with the local perfusion of this antagonist into the mediodorsal thalamic nucleus (MDTN) or into the mPFC showed increased glutamate release after both systemic and local MK-801 administration into the MDTN, but not after local perfusion of MK-801 into the mPFC. Hence, MDTN can be considered as an extremely important brain region generating increased glutamate release following peripheral administration of MK-801. Further detailed analysis of the above-described effects showed that the MK-801-induced blockade of NMDA receptors located on GABAergic neurons in the MDTN led to a reduction in GABA release, and consequently, to a disinhibition of glutamatergic neurons that project into the mPFC and release excessive amounts of glutamate therein. Regarding the role of ARI in modulating the MK-801-evoked glutamate release, these authors suggest that this drug may compensate for hyper-activation of thalamo-cortical glutamatergic neurons via the activation of metabotropic group II glutamate receptors (II-mGluR) localized on their cell bodies in the MDTN and on their axons reaching the mPFC. In support of this suggestion, it was demonstrated that ARI administered locally to both these structures reduced glutamate release in the mPFC induced by systemic administration of MK-801. Similarly to the mode of action of ARI, the study by Fukuyama et al. [94] showed that both systemic NAC administration and local NAC perfusions into the MDTN or mPFC reduced glutamate release in mPFC induced by systemic administration of MK-801. It has also been postulated that NAC-induced activation of cystine-glutamate antiporter in the MDTN and mPFC may contribute to the prevention of the excessive activation of the thalamo-cortical glutamatergic pathway induced by dysfunctional NMDA receptors in the MDTN.

The literature data presented above lead us to suppose that the beneficial effects of both ARI and NAC in reversing social deficits and improving cognitive performance in the BSO-induced neurodevelopmental rat model of schizophrenia may result from the modulating effect of these drugs on the thalamo-cortical glutamatergic transmission. Therefore, further studies are needed to confirm whether the dysfunction of the thalamo-cortical glutamatergic transmission underlies schizophrenia-like social and cognitive deficits in the BSO-induced model of schizophrenia in rats.

In conclusion, it should be mentioned that although the obtained positive effects of the combined ARI and NAC administration in the alleviation negative symptoms and cognitive deficits are promising, the lack of behavioral changes in adult rats pretreated with BSO resembling the positive symptoms commonly observed in patients with schizophrenia may raise some doubts as to the possibility of translating our results to human therapy. Currently, as indicated by the European Psychiatric Association (EPA) to alleviate negative symptoms, it is recommended to administer antidepressants in combination with second-generation neuroleptics [95]. In addition, the EPA guidance group on negative symptoms considers the treatment of depression, positive symptoms and extrapyramidal side effects to be a priority for patients with negative symptoms. Therefore, one might think that it would be more appropriate to use an animal model of schizophrenia expressing positive symptoms to study the influence of ARI and NAC, alone or in combination, on negative symptoms. However, in our recently published studies, which used both neurodevelopmental models of schizophrenia induced by administration of BSO alone or the combination of BSO + GBR 12,909 (with the presence of positive symptoms) [32], chronic administration of ARI (0.3 mg/kg) in combination with an ineffective dose of escitalopram (5 mg/kg), led to an improvement in both social behavior and cognitive functions [96]. The latter results suggest that the two rat models of schizophrenia used are equivalent in terms of assessing the improvement in negative symptoms after administration of an antidepressant in combination with an atypical antipsychotic. In the present study, NAC, which is an antioxidant drug highly recommended for the treatment psychiatric disorders including depression and schizophrenia, was given instead of an antidepressant [97,98]. Adding NAC to ARI therapy in the BSO-induced model of schizophrenia allows for lowering and optimizing the dose of this antipsychotic, which is crucial for treating both negative symptoms and cognitive functions [95]. Finally, in the rat model of schizophrenia induced by the combination of BSO + GBR 2909 (manuscript in preparation), we have made similar observation on the efficacy of ARI and NAC, administered in the same dose range, alone or in combination, in alleviating negative symptoms and improving cognitive function, which indicates the usefulness of both models to assess these symptoms.

## 4. Materials and Methods

The experiments were carried out in compliance with the Act on Experiments on Animals of 21 January 2005 reapproved on 15 January 2015 (published in the Journal of Laws no 23/2015 item 266, Poland), and according to the Directive of the European Parliament and of the Council of Europe 2010/63/EU of 22 September 2010 on the protection of animals used for scientific purposes. The studies received also an approval of the Local Ethics Committee at the Institute of Pharmacology, Polish Academy of Sciences (permission no 3/2018 of January 2018). All efforts were made to minimize the number and suffering of animals used.

### 4.1. Animals and Treatment

To create the neurodevelopmental model of schizophrenia, pregnant Sprague-Dawley females at embryonic day 16 were delivered to our laboratory by the Charles River Company (Sulzfeld, Germany). They were kept in individual cages under standard laboratory conditions; at room temperature (22 °C) under an artificial light/dark cycle (12/12 h), with free access to standard laboratory food and tap water. On the next day after birth, the sex of pups was determined, and only males were left with the dam to be used in further experimental procedure (Figure 5).

Between the postnatal days p5 and p16, male Sprague-Dawley pups were administered the selective inhibitor of γ-glutamate-cysteine ligase (GCL, EC 6.3.2.2.) compound L-buthionine-(S,R)-sulfoximine (BSO, Sigma-Aldrich, Saint Louis, MO, USA) at a dose of 3.8 mmol/kg s.c., once daily, while control pups instead of BSO received a vehicle once daily. BSO was dissolved in 0.9% NaCl. Rats were weighed daily and the injected volumes of the BSO solution was adjusted accordingly to the actual body weight. On postnatal day p23 rats were weaned and housed in groups of four to five until p92 (Figure 5). Sixty-eight-day-old rats that had previously received BSO were chronically treated for 21 consecutive days with N-acetylcysteine (NAC; Sigma-Aldrich, Saint Louis, MO, USA) at doses of 10 or 30 mg/kg orally and/or aripiprazole (ARI; Abcam Biochemicals, Cambridge, UK) administered intraperitoneally at doses of 0.1 or 0.3 mg/kg alone or in combination. In the case of combined therapy, NAC was given 30 min before ARI. NAC was dissolved in 0.9% NaCl, while ARI was dissolved in 0.1 M tartaric acid. The pH of the solution was adjusted to 6–7 with 0.1 N NaOH. The tested drugs (NAC, ARI) were administered in a volume of 2 mL/kg. Control rats, instead of these drugs, were chronically administered 0.9% NaCl solution. The ARI doses used are the same as in our previous studies [60,61], while the NAC doses were selected based on literature data [59]. Behavioral tests evaluating the expression of schizophrenia-like changes corresponding to negative symptoms (social interaction test; SIT) and cognitive deficits (novel object recognition test; NOR), were performed on p90 and p91 days, respectively, in all groups of adult rats, 24 h after the last doses of the tested drugs.

### 4.2. Social Interaction Test

The social interaction test (SIT) was performed using a black PVC box (67 cm × 57 cm × 30 cm, length × width × height). The arena was dimly illuminated with an indirect light of 18 lux [99]. Each social interaction experiment involving two rats was carried out during the light phase of the light/dark cycle. The rats were selected from separate housing cages to make a pair for the study. The paired rats were matched for body weight within 15 g. Each pair of rats was diagonally placed in opposite corners of the box facing away from each other. The behavior of the animals was measured over a 10-min period. The test box was wiped clean between each trial. Social interaction between two rats was expressed as the total time spent in social behavior, such as sniffing, genital investigation, chasing and fighting with each other. The number of episodes was counted as a separate paradigm. Each group was composed of 12 rats (6 pairs).

### 4.3. Novel Object Recognition Test

The novel object recognition (NOR) test was performed using a black PVC box (67 cm × 57 cm × 30 cm, length × width × height).The arena was dimly illuminated with an indirect light of 18 lux. On the first day of the experiment (adaptation), each rat was placed in the box for 10 min. On the next day, the animals was placed in the box for 5 min (T1) with two identical objects (white tin 5 cm wide and 14 cm high or green pyramid 5 cm wide and 14 cm high). The time of object exploration was measured for each of the two objects separately. Then, one hour after T1, the rat was placed again in the box for 5 min (T2), with two different objects: one from the previous session (old) and the other new (white box and green pyramid). The time of object exploration was measured for each of the two objects separately (sniffing, touching or climbing). Each group was composed of 8 rats.

### 4.4. BDNF Expression Analysis

Freshly isolated rat frontal cortex and hippocampus tissues were stored at −80 °C prior to the next analysis. Total RNA was isolated using commercially available Bead-Beat Total RNA Mini Kit (A&A Biotechnology, Gdańsk, Poland) according to the manufacturer’s instructions. After dissolving in water, RNA (1 μg) was reverse-transcribed to cDNA using High Capacity cDNA Reverse Transcription kit with RNAse inhibitor and random hexamers (MultiScribe™, Applied Biosystems, Life Technologies, Carlsbad, CA, USA). The BDNF mRNA level was determined by RealTime PCR using predesigned TaqMan Gene Expression Assays (Applied Biosystems, UK). Assay IDs for the genes examined were as follows: BDNF (Rn01484925_m1) and for reference’s gene HPRT1 (Rn01527840_m1). Amplification was carried out in a total volume of 10 μL (FCx). The mixture containing: 1× FastStart Universal Probe Master (Rox) mix (Roche, Hilden, Germany), 900 nM TaqMan forward and reverse primers and 250 nM of hydrolysis probe labeled with the fluorescent reporter dye FAM at the 5′-end and a quenching dye at the 3′-end and RNAse free water. We used 50 ng of cDNA for the PCR template, Real-Time PCR was conducted using a thermal cycler Quant Studio 3 (Thermo Fisher Scientific, Waltham, MA, USA) and thermal cycling conditions were: 2 min at 50 °C and 10 min at 95 °C followed by 40 cycles at 95 °C for 15 s and at 60 °C for 1 min. Samples were run in duplicate.

### 4.5. ELISA Assay

Freshly isolated rat frontal cortex and hippocampus tissues were stored at −80 °C prior to the analysis. First, the tissues were rinsed in DPBS (GIBCO, Thermo Fisher Scientific, Waltham, MA, USA) to remove excess blood and then they were homogenized in DPBS using Tissue Lyser II (Qiagen Inc., Valencia, CA, USA). The protein measurements of all samples were performed using a BCA Protein Assay Kit (Sigma Aldrich, St. Louis, MO, USA) in pursuance of the manufacturers’ instructions. The protein contents were assessed using a Tecan Infinite 200 Pro spectrophotometer (Tecan, Mannedorf, Germany). Samples containing hippocampal and cortical supernatants were analyzed by enzyme-linked immunoassay (ELISA) using commercially available kits: Rat Brain Derived Neurotrophic Factor ELISA Kit, cat.no. E0476Ra (Bioassay Technology Laboratory, Shanghai, China) according to the manufacturers’ instructions. Briefly, 50 µL of standards and 40 µL of samples, respectively, were dispended into 96 wells coated. Next, 10 µL of anti-BDNF antibodies were added into wells containing samples and 50 µL streptavidin-HRP was added to each well except for blank, and the samples were incubated for 60 min at 37 °C. After washing and next steps as recommended by the manufacturer, the absorbance was determined using a Tecan Infinite 200 Pro spectrophotometer (Tecan, Mannedorf, Germany) set to 450 nm.

### 4.6. Statistics

The Statistical analysis of the obtained results was performed with the use of Statistica version 13.3 for Windows. First, the normality of the data for each studied group were checked using the Shapiro–Wilk test. Then, the Levene’s test was used to check homogeneity of variance. Since all behavioral and biochemical data met both criteria, a one-way analysis of variance (ANOVA) for planned comparisons (the so-called contrast analysis) was applied. The latter analysis was performed in the form of two sets of contrasts. Such a statistical approach enables a fairly reliable estimation of the significance of differences between the selected groups with the smallest possible increase in the α level. The obtained results are presented as the means ± SEM (standard errors of the means) and are considered to be statistically significant when *p* < 0.05. Moreover, in order to demonstrate statistically significant differences between the exploration time of two identical objects by a given rat in the session T1, or between the exploration time of a novel and familiar object by the same rat in the session T2, in each of the tested groups in the NOR test, the Student’s *t*-test for independent samples was used.

## 5. Conclusions

The results of the present study show that addition of N-acetylcysteine to standard therapy with the atypical antipsychotic aripiprazole has a positive effects on amelioration negative symptoms and on improvement of cognitive functions in the BSO-induced rat model of schizophrenia. The absence of positive symptoms in this model of schizophrenia might shed some doubts on the applicability of such therapy in the clinic. However, the combined administration of antipsychotic drugs with non-dopaminergic drugs (antidepressants, N-acetylcysteine) makes it possible to use lower, optimal doses of these drugs and thus, to avoid secondary negative symptoms due to side effects. Hence, the obtained results could also be considered significant in terms of clinical applications.

## Figures and Tables

**Figure 1 ijms-23-02125-f001:**
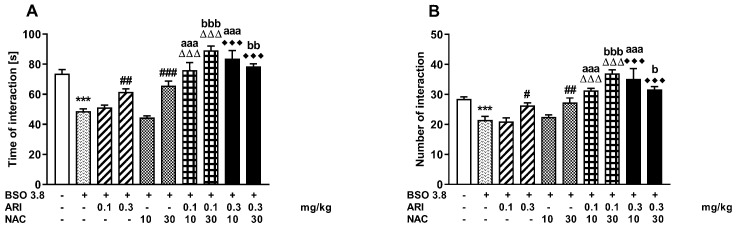
Effects of chronic treatment with aripiprazole (ARI) and N-acetylcysteine (NAC), alone or in combination, on social behavior deficits assessed as the total time spent in social interactions (**A**) and the number of interactions (**B**) in adult Sprague–Dawley rats that were induced by repeated BSO administration in the early postnatal days p5–p16. Data are presented as the mean ± SEM, *n* = 12 (6 pairs) for each group. Statistical analysis was performed using a one-way ANOVA for the planned comparisons; symbols and letters indicate significance of differences between the studied groups, *** *p* < 0.001 vs. control, vehicle-treated group; ^###^ *p* < 0.001, ^##^ *p* < 0.01, ^#^ *p* < 0.05 vs. BSO-treated group; ^ΔΔΔ^ *p* < 0.001 vs. BSO + ARI (0.1 mg/kg)-treated group; ^♦♦♦^ *p* < 0.001 vs. BSO + ARI (0.3 mg/kg)-treated group; ^aaa^ *p* < 0.001 vs. BSO + NAC (10 mg/kg)-treated group; ^bbb^ *p* < 0.001, ^bb^ *p* < 0.01, ^b^ *p* < 0.05 vs. BSO + NAC (30 mg/kg)-treated group.

**Figure 2 ijms-23-02125-f002:**
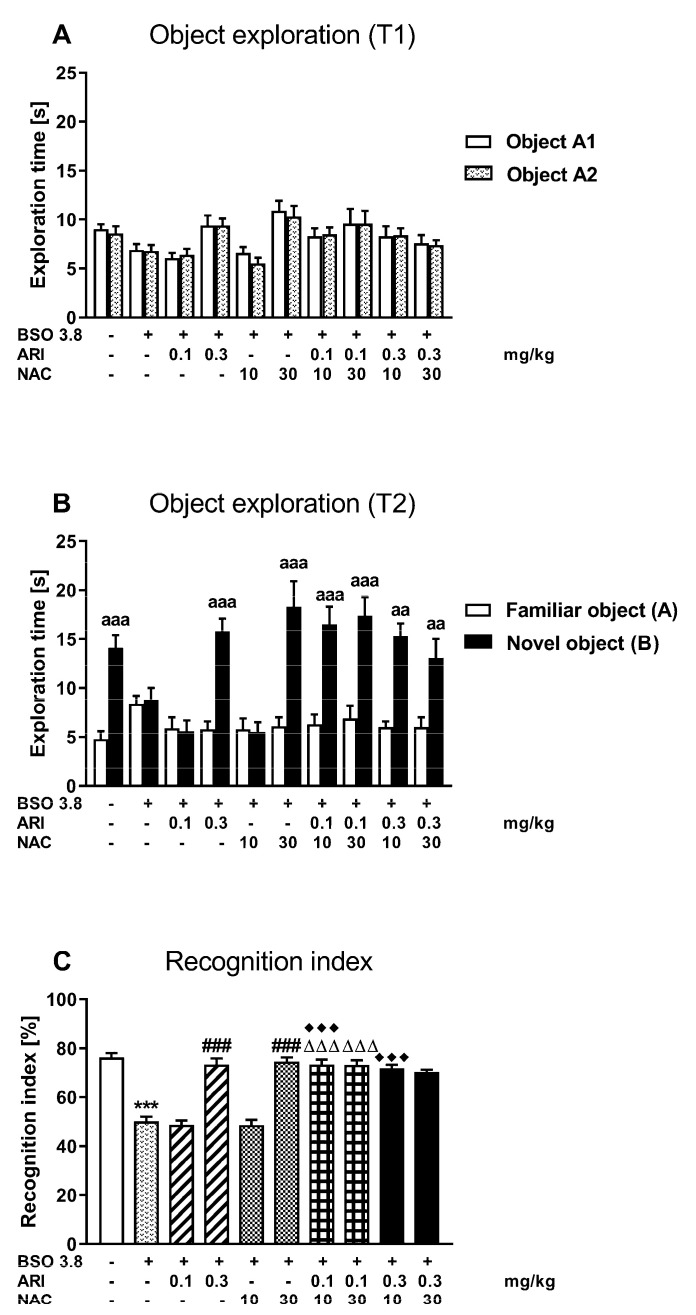
Effects of chronic treatment with aripiprazole (ARI) and N-acetylcysteine (NAC), alone or in combination, on cognitive functions assessed in adult 90-day-old Sprague–Dawley rats that received chronic BSO during early postnatal life (P5–P16). The impact of the studied drugs on: (**A**) exploration of two identical objects in the acquisition trials (session T1), (**B**) exploration of a novel and familiar object in the retention trial (Session T2), (**C**) on the recognition index. Data are presented as the mean ± SEM, *n* = 8 for each group. Letters indicate statistically significant differences between the exploration time of a novel and familiar object in the session T2 within each studied group, according to the Student’s *t*-test for independent samples, ^aaa^ *p* < 0.001, ^aa^ *p* < 0.01 vs. familiar object. Statistical analysis of the recognition index was performed using a one-way ANOVA for the planned comparisons; symbols indicate significance of differences between the tested groups, *** *p* < 0.001 vs. control, vehicle-treated group; ^###^ *p* < 0.001 vs. BSO-treated group; ^ΔΔΔ^ *p* < 0.001 vs. BSO + ARI (0.1 mg/kg)-treated group; ^♦♦♦^ *p* < 0.001 vs. BSO + NAC (10 mg/kg)-treated group.

**Figure 3 ijms-23-02125-f003:**
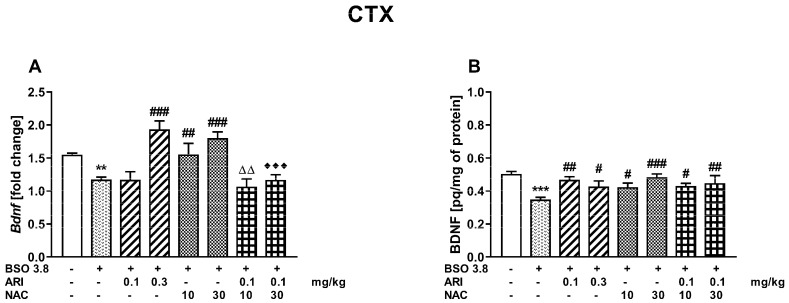
Effects of chronic administration of aripiprazole (ARI) and N-acetylcysteine (NAC), alone or in combination, on BDNF mRNA (**A**) and protein (**B**) levels in the prefrontal cortex of adult 91-day-old Sprague–Dawley rats that received chronic BSO during early postnatal life (P5–P16). Data are presented as the mean ± SEM, *n* = 8 for each group. Statistical analysis was performed using a one-way ANOVA for the planned comparisons; symbols indicate significance of differences between the tested groups, *** *p* < 0.001, ** *p* < 0.01 vs. control vehicle-treated group; ^###^ *p* < 0.001, ^##^ *p* < 0.01, ^#^ *p* < 0.05 vs. BSO-treated group; ^ΔΔ^ *p* < 0.01 vs. BSO + NAC (10 mg/kg)-treated group; ^♦♦♦^ *p* < 0.001 vs. BSO + NAC (30 mg/kg)-treated group.

**Figure 4 ijms-23-02125-f004:**
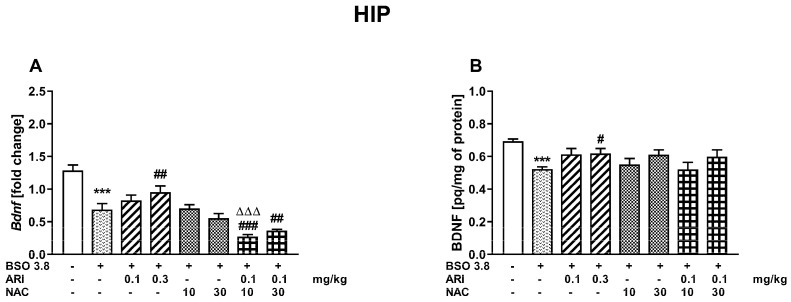
Effects of chronic administration of aripiprazole (ARI) and N-acetylcysteine (NAC), alone or in combination, on BDNF mRNA (**A**) and protein (**B**) levels in the hippocampus of adult 91-day-old Sprague–Dawley rats that received chronic BSO during early postnatal life (P5–P16). Data are presented as the mean ± SEM, *n* = 8 for each group. Statistical analysis was performed using a one-way ANOVA for the planned comparisons; symbols indicate significance of differences between the tested groups, *** *p* < 0.001 vs. control vehicle-treated group; ^###^ *p* < 0.001, ^##^ *p* < 0.01, ^#^ *p* < 0.05 vs. BSO-treated group; ^ΔΔΔ^ *p* < 0.001 vs. BSO + NAC (10 mg/kg)-treated group.

**Figure 5 ijms-23-02125-f005:**
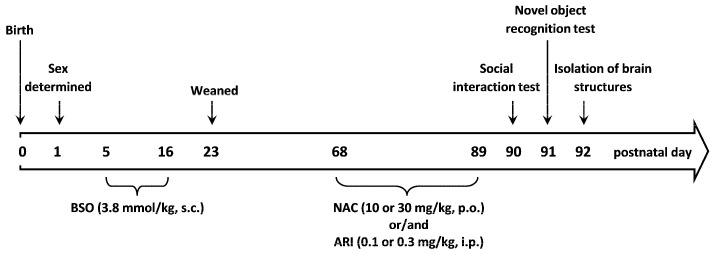
The diagram illustrating the timeline of the experiment.

## Data Availability

Data supporting the reported results are available on request from the corresponding author.

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
