# Peer review of "N-Acetylcysteine and Aripiprazole Improve Social Behavior and Cognition and Modulate Brain BDNF Levels in a Rat Model of Schizophrenia"

_ijms, 2022, doi:10.3390/ijms23042125_

Round 1

Reviewer 1 Report

In this manuscript, Rogóż and colleagues report that a chronic treatment with N-acetylcysteine and aripiprazole reverses schizophrenia-like social and cognitive deficits and perturbations of BDNF levels found in rats exposed to L-buthionine-(S, R)-sulfoximine (BSO) during the early postnatal development. The results are interesting and would be of general interest. However, some points should be addressed before publication:

  • The title is too long and a bit unclear (as well as decreases in the BDNF level is grammatically questionable). The Authors should shorten the title and make it more clear.

  • The discussion is too long as well. Moreover, the Authors discuss a lot about the molecular changes induced by BSO, which have been extensively discussed in their previous studies. Authors need to discuss their results and stay focused on their hypothesis. For instance, the combination of ARI and NAC induces a more powerful effect on social interaction rather than memory impairment. The translational implication of these findings should better discuss. I mean, could the combination of these drugs be more effective in treating social interaction deficits exhibited by schizophrenic patients? Furthermore, the interesting results about BDNF are surprisingly not at all discussed.

  • I suggest to add some statements regarding the potential therapeutic effects of atypical antipsychotics such as ARI on cognitive deficits. Indeed, it has been recently reported that (please report this reference: PMID: 33167370), the ambiguous effects of these drugs on cognitive deficits may depend on genetic variability and that mechanistic advances, which help to understand the heterogeneity of cognitive responses to antipsychotics, may properly guide treatment decisions. These statements will support the rationale of studying the combination of ARI and NAC in this context.

  • The section conclusions is a reiteration of the results. The Authors should discuss the translational value of this work in the conclusions.

  • I also suggest to make some stylistic changes in this manuscript because it appears as a sort of copy of PMID: 34398437.

Author Response

Reviewer's comment 1

Comment 1

The title is too long and a bit unclear (as well as decreases in the BDNF level is grammatically questionable). The Authors should shorten the title and make it more clear.

Reply to comment 1

In accordance with the reviewer's sggestion, the title has been shortened and modified. The current version of the title reads as follows:

N-acetylcysteine and aripiprazole improve social behavior and cognition, and modulate brain BDNF levels in a rat model of schizophrenia

Comment 2

The discussion is too long as well. Moreover, the Authors discuss a lot about the molecular changes induced by BSO, which have been extensively discussed in their previous studies. Authors need to discuss their results and stay focused on their hypothesis. For instance, the combination of ARI and NAC induces a more powerful effect on social interaction rather than memory impairment. The translational implication of these findings should better discuss. I mean, could the combination of these drugs be more effective in treating social interaction deficits exhibited by schizophrenic patients? Furthermore, the interesting results about BDNF are surprisingly not at all discussed.

Reply to comment 2

We have made every effort to tried to shorten the discussion and in fact the second paragraph of the previous version of the discussion has been deleted. As suggested by the reviewer, in the first paragraph of the current version of the manuscript, our attention has been focused on explaining the reasons for the stronger effects of ARI and NAC on negative symptoms than on improving cognitive functions. In the second paragraph, we discussed data referring to BDNF levels in the PFC and HIP. However, in the third section of the discussion, the molecular changes induced by BSO with respect to glutamate levels, which were not previously discussed in any of our earlier works, are retained, as they are important for characterizing this model of schizophrenia.

Comment 3

I suggest to add some statements regarding the potential therapeutic effects of atypical antipsychotics such as ARI on cognitive deficits. Indeed, it has been recently reported that (please report this reference: PMID: 33167370), the ambiguous effects of these drugs on cognitive deficits may depend on genetic variability and that mechanistic advances, which help to understand the heterogeneity of cognitive responses to antipsychotics, may properly guide treatment decisions. These statements will support the rationale of studying the combination of ARI and NAC in this context.

Reply to comment 3

In the first section of the discussion, we have added a few statements regarding potential therapeutic effects of ARI on cognitive deficits. We also include the proposed reference to the list of references cited in our study.

Comment 4

The section conclusions is a reiteration of the results. The Authors should discuss the translational value of this work in the conclusions.

Reply to comment 4

The section conclusions has been changed.

Comment 5

I also suggest to make some stylistic changes in this manuscript because it appears as a sort of copy of PMID: 34398437.

Reply to comment 5

The subject of the present work, due to the model of schizophrenia used, is a continuation of our previous research, therefore, despite all efforts, it is difficult to avoid some stylistic similarities.

Reviewer 2 Report

The present study evaluates the efficacy of aripiprazole and N-acetylcysteine in treating social and cognitive deficits and in increasing brain-derived neurotrophic factor (BDNF) in the prefrontal cortex and hippocampus in a murine model of schizophrenia.

Significant improvements in social cognitive performance as assessed in behavioral tests were observed, as well as specific changes in BDNF mRNA and protein levels.

The study is timely and interesting, the methods are well-described, and the manuscript is well-written and easy to read.

Overall, the paper might represent a valuable contribution to the research field.

Some revisions could further increase the interest and the overall quality of the paper; one methodological issue in particular has to be addressed (see below).

Methods:

-More information regarding the statistical analyses, including the established level of significance and the software used to conduct the analyses, should be provided for better clarity and for the reproducibility of results.

-All post-hoc t-test comparisons have to be corrected with Bonferroni correction. This represents an important methodological issue, and if some results were to change on the basis of this correction they would have to be commented appropriately in the rest of the manuscript.

Discussion:

-More discussion on how these findings could help in the treatment of people living with schizophrenia, perhaps integrated with currently available evidence-based treatment for negative symptoms (see Galderisi S et al. EPA guidance on treatment of negative symptoms in schizophrenia. Eur Psychiatry. 2021;64(1):e21. doi:10.1192/j.eurpsy.2021.13) and cognitive impairment (see Vita A, Barlati S, Ceraso A, et al. Effectiveness, Core Elements, and Moderators of Response of Cognitive Remediation for Schizophrenia: A Systematic Review and Meta-analysis of Randomized Clinical Trials. JAMA Psychiatry. 2021;78(8):848-858. doi:10.1001/jamapsychiatry.2021.0620) could be of interest for the reader.  

-A paragraph explicitly detailing the limitations of the present study is warranted. For instance, the absence in adult rats of symptoms resembling positive symptoms as can be observed in people living with schizophrenia might represent an issue that requires further discussion, as it might raise some doubts on the possibility of extending the result of the study to treatment application in humans.

Author Response

Reviewer's comment 2

Comment 1

Methods:

More information regarding the statistical analyses, including the established level of significance and the software used to conduct the analyses, should be provided for better clarity and for the reproducibility of results.

All post-hoc t-test comparisons have to be corrected with Bonferroni correction. This represents an important methodological issue, and if some results were to change on the basis of this correction they would have to be commented appropriately in the rest of the manuscript.

Reply to comment 1

The incorrect description of the statistical analysis has been removed. The current version of the manuscript presents the correct description of this analysis with all the details. Figure captions relating to one-way ANOVA for the planned comparisons have also been corrected.

Comment 2

Discussion:

More discussion on how these findings could help in the treatment of people living with schizophrenia, perhaps integrated with currently available evidence-based treatment for negative symptoms (see Galderisi S et al. EPA guidance on treatment of negative symptoms in schizophrenia. Eur Psychiatry. 2021;64(1):e21. doi:10.1192/j.eurpsy.2021.13) and cognitive impairment (see Vita A, Barlati S, Ceraso A, et al. Effectiveness, Core Elements, and Moderators of Response of Cognitive Remediation for Schizophrenia: A Systematic Review and Meta-analysis of Randomized Clinical Trials. JAMA Psychiatry. 2021;78(8):848-858. doi:10.1001/jamapsychiatry.2021.0620) could be of interest for the reader.

A paragraph explicitly detailing the limitations of the present study is warranted. For instance, the absence in adult rats of symptoms resembling positive symptoms as can be observed in people living with schizophrenia might represent an issue that requires further discussion, as it might raise some doubts on the possibility of extending the result of the study to treatment application in humans.

Reply to comment 2

The reference to the possibility of extending the obtained results to treatment application in humans was included in the section conclusions. I am grateful for the attached references that helped in formulating these conclusions.

Reviewer 3 Report

This is a well-written research article. It actually provides a baseline for future works on this issue and is highly potential for publication. There are a small number of things that could improve the manuscript. I have outlined these issues below:

  1. Were the paired rats used in social interaction test from the same treatment group? Please clarify it.  
  2. How did social interaction and novel object recognition test be analyzed? Were they done by hand scoring or software detection? Please clarify it.
  3. Please clarify the statistical analysis. Student’s t test post-hoc comparisons was not correct for such ANOVA analysis
  4. Line 202-203: A one-way ANOVA for the planned comparisons performed for the recognition index in all studied groups (F(90,70) = 40.10, P < 0.0001). Please double check the degree of freedom for the numerator of the F ratio.
  5. The number of animals (Line 147: n=12 for each group) used in Figure 1 is different from the number of animals (Line 187: n=8 for each group) used in Figure 2. In addition, the number of animals in Figure 1 and 2 are also different from method section (Line 438, 450). Any exclusion criteria? Please clarify it.
  6. The retention interval between T1 and T2 was chosen for one hour, rather than 24 hr, in the novel object recognition test. One hour retention interval is considered as short-term memory. Does aripiprazole or N-acetylcysteine treatment in BSO model affect long-term memory?

Author Response

Reviewer's comment 3

Comment 1

"This is a well-written research article. It actually provides a baseline for future works on this issue and is highly potential for publication. There are a small number of things that could improve the manuscript. I have outlined these issues below:

Were the paired rats used in social interaction test from the same treatment group? Please clarify it.

How did social interaction and novel object recognition test be analyzed? Were they done by hand scoring or software detection? Please clarify it.

Reply to comment 1

Rats used in the SIT test were housed 4 animals per each separate breeding cage. To perform the test correctly, rats from different cages were paired. The time of social interactions and their number were assessed by two experienced experimenters.

Comment 2

Please clarify the statistical analysis. Student’s t test post-hoc comparisons was not correct for such ANOVA analysis.

Line 202-203: A one-way ANOVA for the planned comparisons performed for the recognition index in all studied groups (F(90,70) = 40.10, P < 0.0001). Please double check the degree of freedom for the numerator of the F ratio.

Reply to comment 2

The incorrect description of the statistical analysis has been removed. The current version of the manuscript presents the correct description of this analysis with all the details. The number of degrees of freedom for the numerator of the F ratio is 9, not 90. This has been corrected.

Comment 3

The number of animals (Line 147: n=12 for each group) used in Figure 1 is different from the number of animals (Line 187: n=8 for each group) used in Figure 2. In addition, the number of animals in Figure 1 and 2 are also different from method section (Line 438, 450). Any exclusion criteria? Please clarify it.

Reply to comment 3

In the SIT test, we examined 12 rats (6 pairs), while in the NOR test only 8 rats from each group were used. There were no exclusion criteria for the animals tested in the NOR test, just 8 rats were randomly selected from the group of 12 rats tested in the SIT test as this number is sufficient to obtain statistically reliable results. In the method section, the number of animals used in the SIT and NOR tests was standardized in accordance with the numbers given in the description of Figures 1 and 2.

Comment 4

The retention interval between T1 and T2 was chosen for one hour, rather than 24 hr, in the novel object recognition test. One hour retention interval is considered as short-term memory. Does aripiprazole or N-acetylcysteine treatment in BSO model affect long-term memory?"

Reply to comment 4

In the present study, we only tested the effects of chronic administration of ARI and NAC, either alone or in combination, on memory 24h after their last dose. Since antipsychotic drugs are given chronically in patients wirth schizophrenia, the aim of the present study was to check long-term effects of these drugs on cognition, but not to check whether these drugs affect a short-term or long-term memory.

Round 2

Reviewer 1 Report

The Authors have addressed all the points I raised. Well done.

Reviewer 2 Report

The statistical analyses section has been rewritten, and is consistently different from the previous one. However, correction for multiple comparisons has again not been applied.

Moreover, the manuscript lacks a paragraph detailing the limitation of the study, which has been explicitly requested in the prevoious revision.